# Comparative Conformational Analysis of Acyclic Sugar Alcohols Ribitol, Xylitol and d-Arabitol by Solution NMR and Molecular Dynamics Simulations

**DOI:** 10.3390/molecules29051072

**Published:** 2024-02-29

**Authors:** Shiho Ohno, Noriyoshi Manabe, Jun Uzawa, Yoshiki Yamaguchi

**Affiliations:** 1Division of Structural Biology, Institute of Molecular Biomembrane and Glycobiology, Tohoku Medical and Pharmaceutical University, 4-4-1 Komatsushima, Aoba-ku, Sendai 981-8558, Miyagi, Japan; s.ohno@tohoku-mpu.ac.jp (S.O.); manabe@tohoku-mpu.ac.jp (N.M.); 2Structural Glycobiology Team, RIKEN (The Institute of Physical and Chemical Research), 2-1 Hirosawa, Wako 351-0198, Saitama, Japan; uzawa_fe79@ck9.so-net.ne.jp

**Keywords:** ribitol, xylitol, arabitol, conformation, dynamics, NMR, MD simulation

## Abstract

Ribitol (C_5_H_12_O_5_) is an acyclic sugar alcohol that was recently identified in *O*-mannose glycan on mammalian α-dystroglycan. The conformation and dynamics of acyclic sugar alcohols such as ribitol are dependent on the stereochemistry of the hydroxyl groups; however, the dynamics are not fully understood. To gain insights into the conformation and dynamics of sugar alcohols, we carried out comparative analyses of ribitol, d-arabitol and xylitol by a crystal structure database search, solution NMR analysis and molecular dynamics (MD) simulations. The crystal structures of the sugar alcohols showed a limited number of conformations, suggesting that only certain stable conformations are prevalent among all possible conformations. The three-bond scholar coupling constants and exchange rates of hydroxyl protons were measured to obtain information on the backbone torsion angle and possible hydrogen bonding of each hydroxyl group. The 100 ns MD simulations indicate that the ribitol backbone has frequent conformational transitions with torsion angles between 180° and ±60°, while d-arabitol and xylitol showed fewer conformational transitions. Taking our experimental and computational data together, it can be concluded that ribitol is more flexible than d-arabitol or xylitol, and the flexibility is at least in part defined by the configuration of the OH groups, which may form intramolecular hydrogen bonds.

## 1. Introduction

Ribitol (C_5_H_12_O_5_) is a sugar alcohol that is a component of teichoic acid in Gram-positive bacteria and of riboflavin (vitamin B2). A recent study showed that ribitol phosphate (d-ribitol 5-phosphate) is also found in the *O*-mannose-type glycan attached onto α-dystroglycan (Figure 1a). Ribitol phosphate in the *O*-mannose glycan is essential for skeletal muscle development, and genetic defects of the enzymes involved in the biosynthesis and transfer of ribitol phosphate lead to muscular dystrophy. These enzymes include fukutin (FKTN), fukutin-related protein (FKRP), and isoprenoid synthase domain-containing protein (ISPD) [1]. The xylose–glucuronic acid (Xyl–GlcA) repeat in *O*-mannose glycan is the interaction site of laminin, and this repeat cannot be built on α-dystroglycan without the ribitol scaffold.

The biosynthesis of tandem ribitol phosphate is achieved by substrate recognition by enzymes including FKTN and FKRP. However, details of the mode of recognition by these enzymes towards such flexible ligands have not been elucidated. Information on the conformation and dynamics of acyclic ribitol will be essential for understanding the structure–function relationships. Acyclic ribitol is different from typical cyclic sugar residues in that the ribitol main chain possesses four rotatable C–C intra-residue linkages (C1–C2, C2–C3, C3–C4 and C4–C5), which potentially give inherent flexibility in addition to the flexibility provided by inter-residue glycosidic linkages. In general, linear compounds were considered to be flexible, and the conformations were not well linked to function. Nonetheless, we have chosen to focus our conformational analysis on linear sugar alcohols. An inspiring paper recently reported on the structure–function relationship of the stereoisomeric compounds, whose biological activities (agonist/antagonist) are dependent on their conformational preferences, either extended or bent [2].

We previously analyzed the conformation and dynamics of ribitol by NMR and molecular dynamics simulations and found asymmetric conformations to be present in ribitol [3]. Furthermore, we found that the conformational dynamics of ribitol are at least in part restricted by the presence of OH groups at C2 and C4, as evidenced by MD simulation of 1,3,5-pentanol, a compound similar to ribitol but lacking OH groups at C2 and C4.

We then turned our attention to other sugar alcohols also composed of five carbons, namely d-arabitol (d-arabinitol), l-arabitol and xylitol (Figure 1b). These have different orientations of OH groups compared with ribitol; d-arabitol differs from ribitol at the C2 position and xylitol at the C3 position. l-arabitol is the enantiomer of d-arabitol. We decided on a comparison of the dynamics of the three sugar alcohols to provide information on how the stereochemistry of each OH group contributes to the conformation and dynamics of sugar alcohols.

In this study, we focus on the three sugar alcohols ribitol, d-arabitol and xylitol, and we compare their conformation and dynamics by database search, solution NMR and MD simulations.

## 2. Results and Discussion

### 2.1. Database Analysis of d-Arabitol and Xylitol

We previously searched the Protein Data Bank (PDB) and the Cambridge Crystallographic Data Centre (CCDC) to extract the conformations of ribitol and ribitol phosphate. Here, we searched for conformations of d-arabitol and xylitol from the CCDC database (Figure 2). We extracted 4 d-arabitol and 45 xylitol conformations, and the dihedral angles of the main chain (φ1, φ2, φ3 and φ4) and the linear C1–C5 distances are listed in Table 1.

The dihedral angles of three ribitol crystals were found to be around 180° for φ1, around −60° for φ2, around 180° for φ3, and around 60° for φ4. The adjacent dihedral angles φ2 and φ3 of ribitol tend to assume different values [3]. The main-chain conformation of ribitol is hence regarded as bent; consequently, the C1–C5 distance of ribitol is short (4.5–4.6 Å). In contrast, d-arabitol crystals showed different conformational properties. d-arabitol tends to show a longer C1–C5 distance (5.1 Å) than ribitol; thus, the conformation of d-arabitol is regarded as extended. The extended conformation of d-arabitol is consistent with the hypothesis of Jeffrey et al. [22], in which the carbon chain of an alditol tends to adopt an extended conformation when the stereo configurations of C_n_ and C_n+2_ are different (e.g., C2 is d and C4 is l or C2 is l and C4 is d as in d-arabitol). To the contrary, when the stereo configurations of C_n_ and C_n+2_ are the same (C2 and C4 are both d for ribitol and xylitol), the parallel orientation of the C_n_–O and C_n+2_–O bonds causes a steric repulsion, and the conformation of the alditol becomes bent [22]. In fact, ribitol and xylitol tend to assume a bent conformation. Xylitol crystals were found to show two conformational patterns, in which φ2/φ3 = 60°/180° (*n* = 33) or 180°/60° (*n* = 12). It has been suggested that xylitol has two patterns of stable conformations [10]. The left- and right-handed bent conformations must be equally probable due to the symmetrical property of xylitol [22]. A structural comparison of these crystal structures indicates that the stable conformation(s) of sugar alcohols are defined by the configuration of the OH bonds.

### 2.2. NMR Analysis of Ribitol and Its Stereoisomers

Solution NMR analysis was performed to investigate the conformation of ribitol and its stereoisomers d-arabitol and xylitol. We assigned the ^1^H- and ^13^C-NMR signals of d-arabitol and xylitol, in addition to those of ribitol [3]. Stereospecific assignment of methylene prochiral protons was performed by comparing experimental and simulated ^1^H-NMR spectra (Figure 3). Due to their symmetrical nature (meso form), H1 to H3 were assigned for ribitol and xylitol (Table 2).

The chemical shifts and the ^2^*J* and ^3^*J* coupling constants are consistent with the reported values for each sugar alcohol [23,24,25]. These coupling constants are used in the Haasnoot equation [26] to estimate the distribution of each conformer. The distribution of the φ1 dihedral angle (180°, −60°, +60°) was calculated using ^3^*J*(H1R,H2)(*pro-R*) and ^3^*J*(H1S,H2)(*pro-S*) based on Hawkes’ method [25]. The φ1 distribution 180°:−60°:+60° was 64:36:0 for ribitol, 57:17:26 for d-arabitol, and 59:21:19 for xylitol (Table 3). These results are almost the same as those of Hawkes [25]. For the dihedral angle φ2 (C1–C2–C3–C4), the conformational population can be estimated from ^3^*J*(H2,H3) and ^3^*J*(C1,H3). The φ2 population 180°:−60°:+60° was 2:46:52 for ribitol and 55:45:0 for xylitol (Table 3). For d-arabitol, the coupling constants corresponding to φ2 could not be obtained due to signal overlap. All three alditols showed a large proportion of 180° for φ1. This is mostly consistent with the crystallographic data shown in Table 1, except in the case of d-arabitol. The population of φ2 was different for ribitol and xylitol. Ribitol φ2 was populated around ±60°, while xylitol φ2 was populated at 180° or −60°. The NMR-derived data on ribitol and xylose (Table 3) are partially consistent with the crystallographic data shown in Table 1. The inconsistencies may arise from differences between crystal and solution states; the former usually include intermolecular hydrogen bonds, which may affect the conformation.

Intramolecular hydrogen bonding is a possible factor affecting the conformation of alditols. To obtain information on hydrogen bonding, we tried to detect OH signals directly (Figure 4). The assignments of the OH signals were deduced from a series of NMR experiments including DQF-COSY, TOCSY and NOESY at 0 °C, and the assignments are labeled for each peak in Figure 4. The chemical shifts and lineshapes of the hydroxyl protons were different for each alditol, suggesting that each OH group is in a different local environment and/or a different hydrogen binding pattern.

At 0 °C, two sharp peaks were observed for ribitol at 5.9 and 5.8 ppm, corresponding to OH3 and OH2/4, respectively. Like for ribitol, two sharp peaks were observed for d-arabitol at 5.9 ppm and 5.6 ppm. They correspond to OH4 and OH3. For xylitol, a single sharp peak was observed at 5.7 ppm, corresponding to OH2/4. These sharp signals are indicative of a relatively slow exchange rate with the solvent and the existence of hydrogen bonding.

To estimate the exchange rates of the hydroxyl protons, a temperature coefficient was calculated. The temperature coefficients were estimated to be 11 ppb·K^−1^ for ribitol OH3, 11 ppb·K^−1^ for xylitol OH2/4 and 10 ppb·K^−1^ for d-arabitol OH4. According to Sandströ et al., the temperature coefficient is greater than 11 ppb·K^−1^ when the OH protons are fully hydrated [27]. This suggests that the hydroxyl group of the alditols is at least partly exposed to the solvent and may not be involved in the formation of stable hydrogen bonds. To quantitatively evaluate the solvent exchange of the OH protons, the exchange rate constant of the OH proton was estimated from 2D EXSY with different mixing times. The exchange rate constants were calculated from the volume of the exchange peak and were estimated to be *k*_ex_ = 193 s^−1^ for ribitol OH3 [3], 810 s^−1^ for OH2/OH4, 908 s^−1^ for d-arabitol OH4 and 1145 s^−1^ for xylitol OH4. Hence, the exchange rates are rather fast, suggesting no stable hydrogen bonding. However, there were significant variations in exchange rates, which may indicate that each hydroxyl proton experiences different hydrogen bonding frequencies.

### 2.3. Conformational Analysis of Alditols by Molecular Dynamics Simulation

MD simulations were then performed for ribitol, d-arabitol and xylitol. The crystal structure of each alditol (ribitol and xylitol) was used as the initial structure, and the simulation time was set to 100 ns. The initial structure of d-arabitol was manually obtained from the mirror image of the l-arabitol crystal structure. For the three alditols, φ1, φ2, φ3 and φ4 were plotted against the simulation time (Simulation #1, Figure 5). To evaluate whether the 100 ns simulation time is enough to search for the stable conformations and their transition, we performed another set of 100 ns MD simulations using different initial structures (Simulation #2, Figure 5). We confirmed that the distribution of each dihedral angle for each alditol was almost the same for #1 and #2 (Table 4). Furthermore, we analyzed the root-mean-square deviation (RMSD) and potential energy of the three alditols in Simulation #1 (Figure 6). The RMSD and the potential energy showed almost uniform values during the simulation. Taking these results together, we concluded that a 100 ns simulation is enough to reach equilibrium.

The stable conformations of each alditol in the MD simulations were similar to those found in CCDC (Table 1). As we discussed in a previous paper [3], φ2 and φ3 of ribitol are correlated; when φ2 is 180°, φ3 tends to occupy 60°, and when φ2 is −60°, φ3 assumes 180°. In d-arabitol, both φ2 and φ3 were mostly found around 180°. This result for d-arabitol is consistent with the CCDC data, and this combination of dihedral angles means an extended structure. Xylitol tended to have φ2 around 180° and φ3 around −60° in the simulations. Contrary to these simulation results, CCDC data put φ2 at both 60° and 180°. A correlation was found for φ3 and φ4 of xylitol; when φ3 is 180°, φ4 tends to occupy −60°, and when φ3 is −60°, φ4 becomes 180°. Taken together, the MD simulations show the correlations between adjacent dihedral angles, φ2 and φ3 in ribitol, and φ3 and φ4 in xylitol.

Jeffrey et al. [22] found that the carbon chain of alditol tends to adopt an elongated zigzag conformation when the stereo configuration of C_n_ and C_n+2_ is different, as in the case of d-arabitol (C2 and C4 are l and d). In contrast, ribitol and xylitol have the same configuration of C2 and C4, namely d and d, and the zigzag extended conformation causes steric hindrance. Therefore, the overall conformations of ribitol and xylitol tend to become bent. The probability of both φ2 and φ3 being 180° simultaneously was 36% for ribitol, 42% for xylitol, and 82% for d-arabitol in the MD simulations. To define the bent/extended conformations for each alditol, we plotted the linear C1–C5 distance obtained during the simulations (Figure 7). It was found that ribitol frequently flipped between extended (5.3 Å) and bent (4.5 Å). d-arabitol and xylitol showed similar transitions, but far less frequently. It can be concluded that all sugars undergo the extended/bent transitions, and ribitol is the most dynamic in this regard.

The intramolecular hydrogen bonding during the simulations was analyzed to examine the effect of hydrogen bonding on the conformation and dynamics. The formation of intramolecular hydrogen bonds of each alditol was analyzed, and those with more than 10% probability are shown in Figure 8; the frequency of hydrogen bond formation is shown for each oxygen in Table 5.

When the cutoff was set at 20%, the hydrogen-bonding OH groups were OH2, OH3 and OH4 for ribitol, OH3–OH5 for d-arabitol, and OH2 and OH4 for xylitol. In general, these results are consistent with our NMR analysis of OH protons. From these data, it can be concluded that the zigzag extended conformation is the more favored one for alditols; however, the steric hindrance of the parallel OH arrangement and intramolecular hydrogen bonding are contributory factors regulating the stability and dynamics.

In summary, the conformation and dynamics of three acyclic sugar alcohols—ribitol, d-arabitol and xylitol—were examined by database search, NMR experiments and MD simulations. The configuration of the OH groups affects the conformation and dynamics, and adjacent dihedral angles are mutually correlated with each other: φ2 and φ3 of ribitol and φ3 and φ4 of xylitol. This correlation was most evident in MD simulations. Ribitol is more flexible than d-arabitol or xylitol, and this characteristic of ribitol may be important for efficient laminin–α-dystroglycan interaction.

Our methodology is basically applicable to other molecules including cyclic sugars. Molecular dynamics simulations are applicable for larger systems such as glycans, proteins and other biopolymers with particular attention to the force field and simulation time. The experimental approach, especially NMR, is greatly affected by the molecular size, and the method needs optimization such as isotope labeling to be applied to larger systems.

## 3. Materials and Methods

### 3.1. Database Analysis

The three-dimensional (3D) crystal structures of ribitol, d-arabitol and xylitol were extracted from The Cambridge Crystallographic Data Centre (CCDC, as of October 2023). The dihedral angles of ribitol, d-arabitol and xylitol were defined as φ1 (O1–C1–C2–C3), φ2 (C1–C2–C3–C4), φ3 (C2–C3–C4–C5) and φ4 (C3–C4–C5–O5) and measured using Discovery Studio.

### 3.2. MD Simulations

MD simulations were performed based on GENESIS 1.7.0 [28,29]. The coordinates of ribitol and xylitol were extracted from the PDB database (PDB ID: 5IAI, 5Y4J, 4Q0S, 4RS3) and were used as the initial structure. CHARMM36 was assigned as the force field [30]. The coordinates of d-arabitol were manually prepared from the coordinates of l-arabitol in the PDB database (PDB ID: 4R1Q) by changing the chirality of each carbon. Hydrogens and explicit water molecules were generated using the “Solution Builder” protocol CHARMM-GUI (https://www.charmm-gui.org/, accessed on 1 August 2022). The simulation time was set to 100 ns. Explicit water molecules were placed for each coordinate of ribitol, d-arabitol and xylitol (9357–9817 water molecules), and TIP3 was used as the water model [31]. The size of the simulation system was 67–68 Å × 67–68 Å × 67–68 Å, which guarantees that the distance between periodic images of the molecules is 30 Å. A 10,000-step minimization was performed, which eliminated distortion of the entire structure with the steepest descent algorithm. Heating was performed under an *NVT* ensemble at 300 K. After heating, production was performed under an *NPT* ensemble with a time step of 2 fs. The number of hydrogen bond dihedral angles of each conformer was calculated using GENESIS 1.7.0. Hydrogen bonds in GENESIS software were defined as meeting the following three criteria: (1) donor and acceptor oxygens are within 3.4 Å; (2) the donor hydrogen–donor oxygen–acceptor oxygen formation angle is within 30°; and (3) the donor oxygen–donor hydrogen–acceptor oxygen formation angle is greater than 120°. The RMSD and potential energy for each alditol were also calculated using GENESIS software. The RMSD was calculated using all atoms in the compound.

### 3.3. Solution NMR Analysis

Ribitol and d-arabitol were purchased from Tokyo Chemical Industry (Tokyo, Japan). Xylitol was purchased from nacalai tesque (Kyoto, Japan). Solution NMR analyses were performed using JNM-ECZ600R/S1 (JEOL, Tokyo, Japan). The probe temperature was set from 273 to 303 K. A sample (18–20 mg) was dissolved in 600 μL of D_2_O for signal assignment or 10 mM sodium acetate buffer, pH 6.0 (H_2_O:D_2_O = 1:1) for the detection of OH signals. ^1^H and ^13^C chemical shifts were reportedly related to the internal standard of 4,4-dimethyl-4-silapentane-1-sulfonic acid (DSS, 0 ppm). NMR chemical shifts of alditols were assigned by analyzing the 1D-^1^H, 1D-^13^C, 2D-DQF-COSY, NOESY, TOCSY, ^1^H-^13^C HSQC and ^1^H-^13^C HMBC spectra. The ^3^*J*(H,H) and ^2^*J*(H,H) coupling constants of alditols were obtained by comparison of the simulated NMR spectra. ^3^*J*(C,H) coupling constants were obtained from the HR-HMBC spectra [32]. A scaling factor (*n*) of 25 was used, and the digital resolutions for f1 (^13^C) and f2 (^1^H) were 4.4 and 0.7 Hz/point, respectively. The conversion from the coupling constant ^3^*J*(H,H) to the dihedral angle was performed using the Haasnoot equation, which includes a correction for the electronegativity of substituents and the orientation of each substituent relative to the coupled protons [26]. ^3^*J*(C,H) was also applied to an empirical prediction equation [33,34]. The exchange rate of hydroxyl protons with water was calculated from 2D chemical exchange spectroscopy [35,36]. Mixing times of 3 to 24 ms with increments of 3 ms were used. The exchange rate constant was calculated from the initial build-up rate of the diagonal peak volume over the exchange cross-peak. The temperature coefficient (ppb·K^−1^) of hydroxyl protons was measured by collecting a series of 1D-^1^H-NMR spectra at different temperatures (273, 278, 283, 288, 293, 298 and 303 K). NMR data processing was performed using Delta5. 3. 1 (JEOL, Tokyo, Japan), and NMR spectral analyses were performed using Mnova (Mestrelab Research, Santiago, Spain).

## Figures and Tables

**Figure 1 molecules-29-01072-f001:**
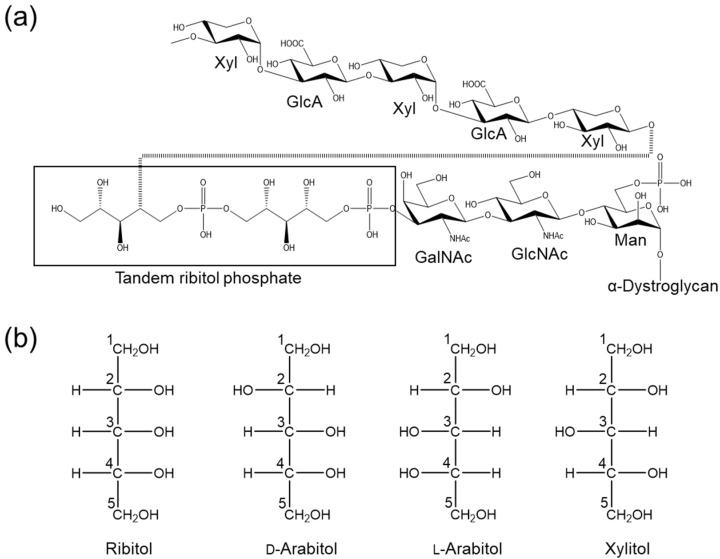
Ribitol residues in *O*-mannose-type glycan and ribitol-related sugar alcohols. (**a**) *O*-Mannose-type glycan containing mannose (Man), GlcNAc, GalNAc, two ribitol phosphates and xylose–glucuronic acid (Xyl–GlcA) repeats which are responsible for laminin binding. (**b**) Ribitol and its related stereoisomers. From left to right: ribitol, d-arabitol, l-arabitol and xylitol.

**Figure 2 molecules-29-01072-f002:**
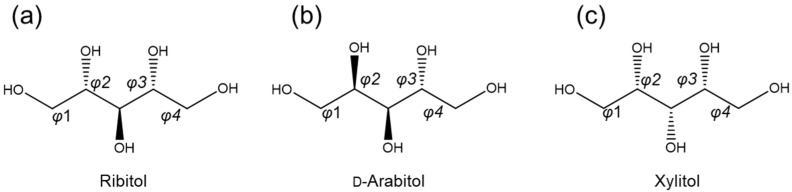
Chemical structure of ribitol and its stereoisomers: (**a**) ribitol, (**b**) d-arabitol and (**c**) xylitol. The main-chain dihedral angles are defined as φ1 (O1–C1–C2–C3), φ2 (C1–C2–C3–C4), φ3 (C2–C3–C4–C5) and φ4 (C3–C4–C5–O5) in this manuscript.

**Figure 3 molecules-29-01072-f003:**
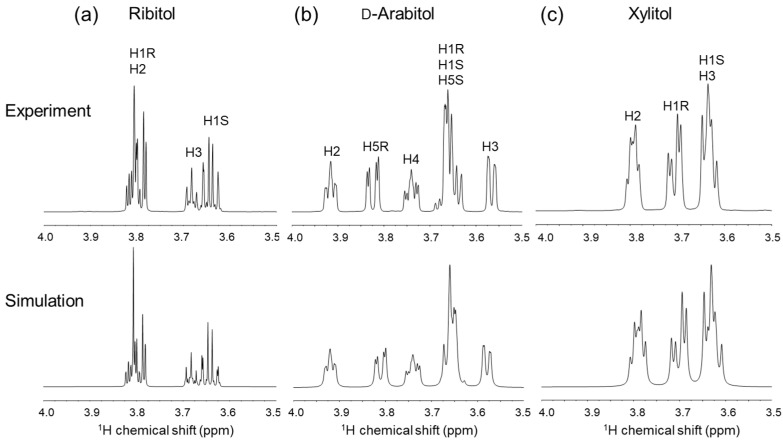
Experimental and simulated ^1^H-NMR spectra of ribitol, d-arabitol and xylitol: (**a**) ribitol [3], (**b**) d-arabitol, and (**c**) xylitol (3.5~4.0 ppm). Upper panels are the experimental data, and the lower panels are the corresponding simulated spectra. The NMR data were obtained at ^1^H observation frequency of 600 MHz. Assignments are labeled on each peak.

**Figure 4 molecules-29-01072-f004:**
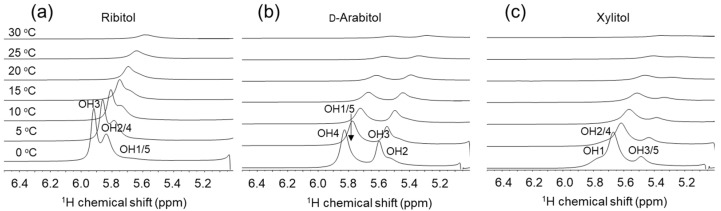
Part of the 1D-^1^H-NMR (OH region) spectra of ribitol [3], d-arabitol and xylitol measured at 0, 5, 10, 15, 20, 25 and 30 °C in 10 mM sodium acetate buffer, pH 6.0 (H_2_O:D_2_O = 1:1): (**a**) ribitol, (**b**) d-arabitol and (**c**) xylitol.

**Figure 5 molecules-29-01072-f005:**
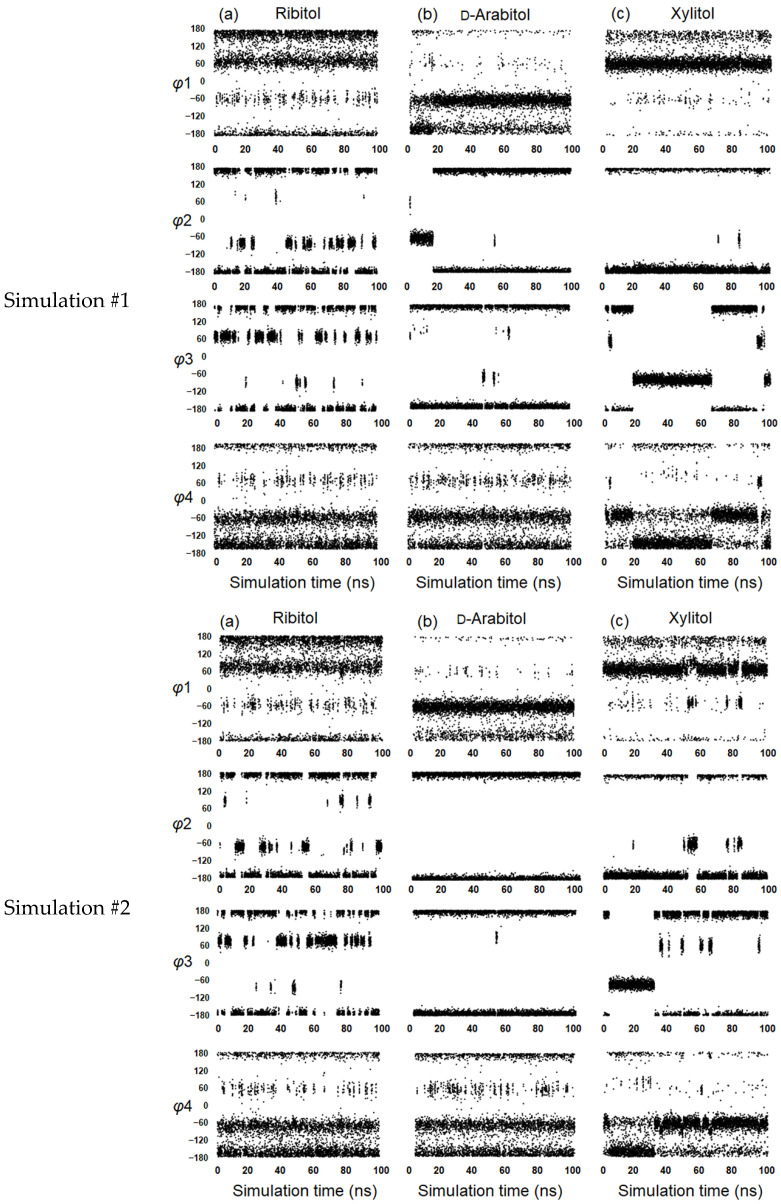
Dihedral angle distribution of each alditol in MD simulations #1 and #2: (**a**) ribitol, (**b**) d-arabitol and (**c**) xylitol. The dihedral angles are defined as φ1 (O1–C1–C2–C3), φ2 (C1–C2–C3–C4), φ3 (C2–C3–C4–C5) and φ4 (C3–C4–C5–O5).

**Figure 6 molecules-29-01072-f006:**
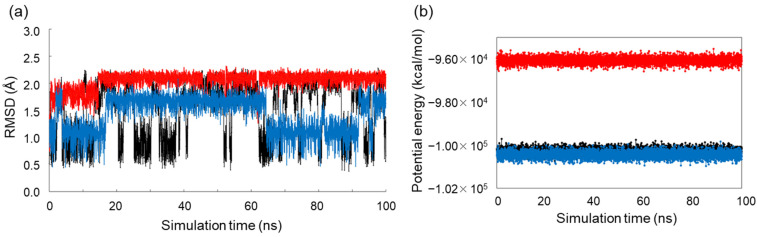
Root-mean-square deviation (RMSD) and potential energy of each alditol in 100 ns MD Simulation #1. (**a**) RMSD of each alditol. Calculation of the RMSD was performed using all atoms of alditols. (**b**) Potential energy of each alditol. In (**a**,**b**), ribitol is shown in black, d-arabitol in red and xylitol in blue.

**Figure 7 molecules-29-01072-f007:**
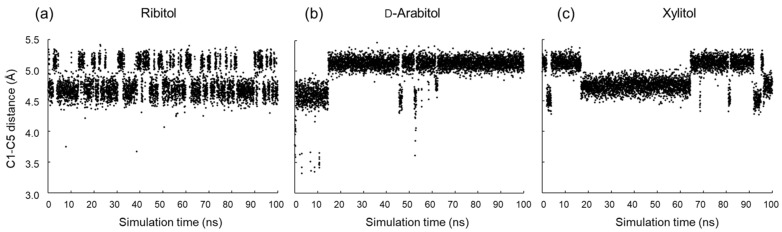
Linear C1–C5 distance of each alditol during MD simulation: (**a**) ribitol, (**b**) d-arabitol and (**c**) xylitol.

**Figure 8 molecules-29-01072-f008:**
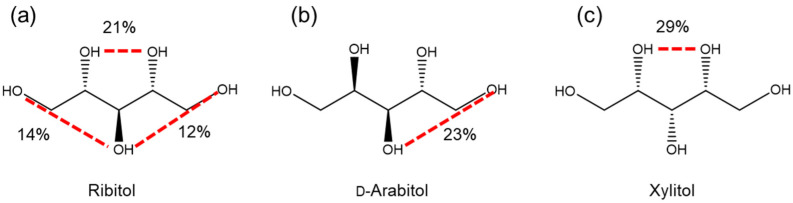
Hydrogen bonding sites and their frequencies for each compound, showing sites with bonding frequencies of more than 10%: (**a**) ribitol, (**b**) d-arabitol and (**c**) xylitol.

**Table 1 molecules-29-01072-t001:** Main-chain dihedral angles (°) and C1–C5 distance (Å) of ribitol [3], d-arabitol and xylitol extracted from the CCDC.

CCDC Deposition Number	φ1	φ2	φ3	φ4	C1–C5 Distance	References
Ribitol [3]						
1249410	171	−62	−172	71	4.5	[4]
662559	171	−61	−171	73	4.6	[5]
1015979	171	−61	−170	73	4.6	[6]
d-Arabitol						
1103512	58	178	178	64	5.1	[7]
1287063	173	174	176	66	5.1	[8]
2150172	177	175	180	62	5.1	[9]
2076894	64	176	171	56	5.1	[10]
Xylitol						
1298203	173	70	176	175	4.5	[11]
223330	173	70	176	175	4.5	[12]
1015980	173	70	176	175	4.5	[6]
1432562	173	70	176	175	4.5	[13]
1423912	173	70	176	175	4.5	[14]
2008303	173	70	176	175	4.5	[15]
2021790	173	70	177	175	4.5	[16]
2021791	175	176	70	174	4.5	[16]
2021792	176	176	69	175	4.4	[16]
2021793	175	177	70	176	4.4	[16]
2021794	175	176	70	174	4.4	[16]
2021795	175	176	71	177	4.4	[16]
2021796	175	174	69	179	4.4	[16]
2021798	178	163	72	179	4.4	[16]
2021799	175	175	69	177	4.4	[16]
2021800	172	173	69	173	4.4	[16]
1898654	175	176	70	173	4.5	[17] *
2008299	173	70	176	175	4.5	[15]
2008300	173	70	176	175	4.5	[15]
2008301	173	70	176	175	4.5	[15]
2008302	173	70	176	175	4.5	[15]
2008304	173	70	176	175	4.5	[15]
2008266	173	70	176	175	4.5	[15]
2008270	173	70	176	175	4.5	[15]
2008283	173	70	176	175	4.5	[15]
2008280	173	70	176	175	4.5	[15]
2060247	173	70	176	175	4.5	[18]
2085455	173	70	176	175	4.5	[15]
1987830	173	70	176	175	4.5	[18,19]
1962229	173	70	176	175	4.5	[20]
1962230	173	70	176	175	4.5	[20]
1962231	173	70	176	175	4.5	[20]
1962232	173	70	176	175	4.5	[20]
1984051	173	70	176	175	4.5	[20]
2081791	173	70	176	175	4.5	[21]
2081792	174	70	176	175	4.5	[21]
2081793	174	70	176	175	4.5	[21]
2081794	173	70	176	175	4.5	[21]
2076896	175	176	70	173	4.5	[10]
2076897	173	70	176	175	4.5	[10]
2008298	173	70	176	175	4.5	[15]
2021797	172	174	73	180	4.4	[16]
2081788	173	70	176	175	4.5	[21]
2081789	173	70	176	175	4.5	[21]
2081790	173	70	176	175	4.5	[21]

* [17] is an online experimental database.

**Table 2 molecules-29-01072-t002:** ^1^H and ^13^C chemical shifts and coupling constants of ribitol, d-arabitol [3] and xylitol determined in this study.

Chemical Shift (ppm)	Ribitol [3]	d-Arabitol	Xylitol
H1R (*pro*-*R*)	3.79	3.66	3.70
H1S (*pro*-*S*)	3.64	3.66	3.63
H2	3.81	3.92	3.79
H3	3.68	3.58	3.63
H4	-	3.74	-
H5R (*pro*-*R*)	-	3.81	-
H5S (*pro*-*S*)	-	3.66	-
C1	65.1	65.8	65.3
C2	74.8	73.0	74.6
C3	74.9	73.2	73.5
C4	-	73.7	-
C5	-	65.7	-
Coupling constant (Hz)			
^3^*J*(H1R,H2)	3.00	5.00	4.5
^3^*J*(H1S,H2)	7.20	7.55	7.5
^3^*J*(H2,H3)	6.50	2.00	4.6
^3^*J*(H3,H4)	-	8.4	-
^3^*J*(H4,H5R)	-	3.05	-
^3^*J*(H4,H5S)	-	6.50	-
^2^*J*(H1R,H1S)	−12.70	−11.55	−11.7
^2^*J*(H5R,H5S)	-	−11.00	-
^3^*J*(C1,H3)	3.8 *	ND	5.4 *
^3^*J*(C3,H1S)	2.9 *	ND	-
^3^*J*(C3,H1R)	ND	ND	4.5 *
^3^*J*(C3,H5)	ND	ND	-
^3^*J*(C5,H2)	ND	ND	10.0 *

* ^3^*J*(C,H) may include errors due to the presence of strong ^3^*J*(H,H) coupling.

**Table 3 molecules-29-01072-t003:** Population of φ1 and φ2 conformations for ribitol [3], d-arabitol and xylitol calculated based on coupling constants.

	φ1			φ2		
	180°	−60°	+60°	180°	−60°	+60°
Ribitol	64	36	0	2	46	52
d-Arabitol	57	17	26	N.D.	N.D.	N.D.
Xylitol	59	21	19	55	45	0

N.D., not determined due to signal overlap.

**Table 4 molecules-29-01072-t004:** The population of each dihedral angle of ribitol, d-arabitol and xylitol from the 100 ns MD Simulations #1 and #2. Each dihedral angle was classified into three conformations: 180° (<−120° and >120°), −60° (−120° to 0°) and +60° (0° to 120°). The population obtained from Simulation #2 is shown in parentheses.

Ribitol	φ1	φ2	φ3	φ4
180°	53 (52)	72 (74)	58 (52)	53 (53)
60°	38 (38)	1 (6)	38 (44)	8 (7)
−60°	9 (10)	27 (21)	4 (3)	39 (40)
d-Arabitol				
180°	26 (18)	85 (100)	96 (99)	44 (49)
60°	2 (2)	0 (0)	2 (1)	12 (11)
−60°	72 (80)	15 (0)	2 (0)	45 (41)
Xylitol				
180°	16 (17)	99 (89)	43 (64)	49 (43)
60°	81 (76)	0 (0)	5 (9)	4 (3)
−60°	3 (7)	1 (11)	51 (28)	47 (63)

**Table 5 molecules-29-01072-t005:** Probability of hydrogen bond formation for each hydroxyl group (%).

	Ribitol	d-Arabitol	Xylitol
O1	14	26	2
O2	22	3	30
O3	26	23	7
O4	22	10	29
O5	12	9	6

## Data Availability

The data presented in this study are available in this article.

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
