# Peer review of "Comparative Conformational Analysis of Acyclic Sugar Alcohols Ribitol, Xylitol and d-Arabitol by Solution NMR and Molecular Dynamics Simulations"

_molecules, 2024, doi:10.3390/molecules29051072_

Round 1
Reviewer 1 Report
Comments and Suggestions for Authors
Comment:
The study titled "Comparative Conformational Analysis of Acyclic Sugar Alcohols Ribitol, Xylitol, and D-Arabitol by Solution NMR and Molecular Dynamics Simulations" explores the conformation and dynamics of acyclic sugar alcohols, focusing on ribitol found in mammalian α-dystroglycan. Through crystal structure analysis, solution NMR, and 100-ns molecular dynamics simulations, the authors compared ribitol with D-arabitol and xylitol. Crystal structures revealed limited conformations, indicating stable prevalent forms. Experimental data, including three-bond scholar coupling constants and exchange rates, provided insights into backbone torsion angles and hydrogen bonding. Molecular dynamics (MD) simulations highlighted ribitol's greater flexibility, attributed to frequent conformational transitions influenced by hydroxyl group configurations. Overall, the work combines experimental and computational approaches to understand the conformational dynamics of acyclic sugar alcohols. The manuscript is well-organized and clearly stated. I would suggest accepting it after the following concerns are addressed.
Major points:
1. In lines 166-171, the authors wrote that “To evaluate whether the 100-ns simulation time is enough to search for the stable conformations and their transition, we performed another set of 100-ns MD simulation using different initial structures (Simulation #2, Figure 5). We confirmed that the distribution of each dihedral angle for each alditiol was almost the same for #1 and #2 (Table 4). Therefore, we concluded that a 100-ns simulation is enough to reach equilibrium independent of initial structures.” The traditional method for evaluating the stability and equilibration of molecular dynamics simulations often involves analyzing the Root Mean Square Deviation (RMSD) of the structures. The RMSD provides a measure of the deviation between the structures at different time points from a reference structure. A stable simulation would show a convergence of RMSD values. While the approach mentioned in this manuscript, comparing the distribution of dihedral angles for different initial structures, can offer insights into the consistency of the simulation results, it may not fully capture the overall stability and equilibration of the system. To ensure a comprehensive assessment of stability, it is recommended to include RMSD analysis along with other relevant analyses, such as Root Mean Square Fluctuation (RMSF), potential energy, and structural clustering. These analyses collectively provide a more robust understanding of the system's behavior during the simulation and its convergence to equilibrium.
Minor points:
1. In line 198, “Cn+2” should be “Cn2+”.
2. In line 255, the 'NPT ensemble,' 'NPT' is typically represented in uppercase letters. Therefore, the correct abbreviation is 'NPT,' not 'nPT.'
3. In lines 255-260, the authors wrote that “The number of hydrogen bond dihedral angles of each conformer was calculated using GEN-ESIS 1.7.0. Hydrogen bond definitions were defined as meeting the following three criteria; 1. Donor and acceptor oxygens are within 3.4 Å; 2. Donor hydrogen-donor oxygen-acceptor oxygen formation angle is within 30°; 3. Donor oxygen-donor hydrogen-acceptor oxygen formation angle is greater than 120°.” Specific studies or software tools may have variations in the criteria based on their goals or methods. based on https://www.r-ccs.riken.jp/labs/cbrt/samples/hb_analysis/, I suggest that authors modify the “Hydrogen bond definition were…” as “Hydrogen bond definitions in GEN-ESIS software were…”.
4. The ref.16 form does not include the article title and journal information. Additionally, I cannot read the ref.16 by “Kacper Pajak, M.M. 2020, doi:10.5517/ccdc.csd.cc21qpxd. ”
Comments on the Quality of English Language
English in full manuscript is good.
Author Response
Our point-by-point responses are uploaded as a word file.

Reviewer 2 Report
Comments and Suggestions for Authors
In this manuscript, authors summarize the conformation and dynamics of three acyclic sugar alcohols, ribitol, D-arabitol and xylitol were examined by database search, NMR experiments and MD simulations. The configuration of the OH groups affect the conformation and dynamics, and adjacent dihedral angles are mutually correlated with each other: φ2 and φ3 of ribitol, φ3 and φ4 of D-arabitol and φ3 and φ4 of xylitol. This correlation was most evident in MD simulations. Ribitol is more flexible than D-arabitol or xylitol and this characteristic of ribitol may be important for efficient laminin-α-dystroglycan interaction. However, there are some concerns that needs be addressed and the following modifications are required before the final acceptance.
1. In the “Introduction” of the manuscript, the authors indicate that “information on the conformation and dynamics of acyclic ribitol is essential for understanding the structure-function relationship of such a sugar alcohol”. Could the authors clarify the reasons?
2. Some figures in the manuscript are marked with (a), (b), (c), etc., while others are not marked. To improve the reading experience of readers and maintain consistency in the format of manuscript figures, it is recommended to add markings such as (a), (b), (c), etc. in the figures to clearly identify each section.
3. Regarding the “Materials and Methods”, describe that “the distance between periodic images of the molecules is 12 Å” and " Hydrogen bond definitions were defined as meeting the following three criteria: 1. Donor and acceptor oxygens are within 3.4 Å; 2. Donor hydrogen-donor oxygen-acceptor oxygen formation angle is within 30°; 3. Donor oxygen-donor hydrogen-acceptor oxygen formation angle is greater than 120°". How are these involved parameter values obtained? Please provide reasonable explanations for the above description.
4. Authors should check the consistency and normalization of the format for the references. For example, there are two "DOI" formats in the manuscript.
Author Response

(The authors gave the same response as above.)

Reviewer 3 Report
Comments and Suggestions for Authors
This is a scientifically solid article describing studies of the conformational properties of several acyclic sugar alcohols.
The studies were done with a combined experimental and computational approach, which give added strength to the presented conclusions.
The conformational flexibility was found to be dependent on the formation of the intramolecular hydrogen bonds.
This work is significant both in the applied and methodological aspects. First, it presents a workable technique that can be applied to other systems. Second, the technique is applied to a specific set of systems of interest.
The article is written clearly and is easy to follow. The data and methods are presented concisely but in sufficient detail.
In the opinion of this reviewer, the manuscript should be publishable following the minor revisions outlined below:
(1) It would be very beneficial to further describe the reason for choosing the specific systems of study and their connection to the broader scope of related problems. Once again, the importance of the particular choice of the study systems should be justified further.
(2) The applicability of the presented techniques and study protocol to other systems of interest should be discussed more explicitly. For instance, could they be used for cyclic systems? Would there be a way to extend the presented methodology to proteins and/or other polymers?
Once again, this is a good manuscript overall, and it should be publishable once the above questions have been addressed.
Author Response

(The authors gave the same response as above.)

Round 2
Reviewer 1 Report
Comments and Suggestions for Authors
The current PDF version appears to have some issues, as I am seeing an overlap between Table 4 and Figure 6 on the pages. However, I acknowledge the results of this work and suggest making slight modifications before accepting it.